# Energy-Effective Data Gathering for UAV-Aided Wireless Sensor Networks

**DOI:** 10.3390/s19112506

**Published:** 2019-05-31

**Authors:** Bin Liu, Hongbo Zhu

**Affiliations:** 1Jiangsu Key Laboratory of Wireless Communications, Nanjing University of Posts and Telecommunications, Nanjing 210003, China; 1015010331@njupt.edu.cn; 2Engineering Research Center of Health Service System Based on Ubiquitous Wireless Networks, Ministry of Education, Nanjing University of Posts and Telecommunications, Nanjing 210003, China

**Keywords:** unmanned aerial vehicles (UAVs), wireless sensor networks, recursive random search (RRS), trajectory optimization, dynamic programming

## Abstract

Unmanned aerial vehicles (UAVs) are capable of serving as a data collector for wireless sensor networks (WSNs). In this paper, we investigate an energy-effective data gathering approach in UAV-aided WSNs, where each sensor node (SN) dynamically chooses the transmission modes, i.e., (1) waiting, (2) conventional sink node transmission, (3) uploading to UAV, to transmit sensory data within a given time. By jointly considering the SN’s transmission policy and UAV trajectory optimization, we aim to minimize the transmission energy consumption of the SNs and ensure all sensory data completed collected within the given time. We take a two-step iterative approach and decouple the SN’s transmission design and UAV trajectory optimization process. First, we design the optimal SNs transmission mode policy with preplanned UAV trajectory. A dynamic programming (DP) algorithm is proposed to obtain the optimal transmission policy. Then, with the fixed transmission policy, we optimize the UAV’s trajectory from the preplanned trace with recursive random search (RRS) algorithm. Numerical results show that the proposed scheme achieves significant energy savings gain over the benchmark schemes.

## 1. Introduction

Wireless sensor networks have been largely deployed for various sensory solutions: agricultural and environmental monitoring, or smart cities application [1,2,3], enabled by an enormous amount of sensor nodes (SNs). The sensory nodes typically send the collected data to the sink node. Constrained by the distance and channel character, the direct sink node transmission approach is nevertheless energy-consuming. Powered by battery, these sensor nodes are difficult to be recharged [4]. Hence, It is of great importance to design the energy-efficient data collection approach for prolonging the lifetime of wireless sensor networks (WSNs) [5]. The mobile sink was considered in [6] to enhance the network performance in the WSNs.

Recently, utilizing the unmanned aerial vehicle (UAV) for WSNs has been deemed as a promising solution of energy-efficient data collection. Featured by flexible mobility, UAV can move sufficiently close to the SNs. The line-of-sight (LoS) communication links between UAV and SNs can reduce transmission energy consumption.

Different from conventional communication techniques, UAV’s deployment and trajectory optimization issues are the new challenges for UAV-aided wireless communications. The performance is enhanced by the UAV placement optimization [7]. Many efforts have been devoted to the UAV’s deployment optimization. Aiming at downlink coverage maximization, a three-dimensional (3D) UAV deployment optimization was considered in [8]. Moreover, 3D UAV deployment was also investigated in [9] to maximize the serving users. Apart from the placement optimization, the authors in [10] maximized throughput by power allocation together with the UAV’s trajectory optimization. The compressive sensing (CS) for WSN data gathering was studied in [11] to offer a transmission-efficient architecture. Using the UAV as LoRaWAN gateways, an energy-efficient surveillance scheme was investigated in [12] for the intelligent transportation systems (ITS).

As a promising solution for data collection and dissemination, UAV-aided WSNs has captured great attention. With practical protocols and experiments, characterization of communication links of UAV-aided WSNs was given in [13]. The multi-sensor uplink interference mitigation was proposed in [14] by leveraging the UAV beamforming. The SNs’ wake-up policy and UAV’s trajectory were jointly optimized in [5] to minimize energy consumption of SNs. The UAV flight time for sensory data collection was minimized in [15]. The multi-agent reinforcement learning (MARL) framework was proposed in [16], where each UAV acts as the agent and automatically selected its communicating node, power levels and subchannels without any information exchange among UAVs. Leveraging proactive caching, reference [17] disseminated the popular data to a set of selected nodes that cooperatively cache all the files. Then, one SN retrieved the requested data from its local cache or from its nearest neighbor that has cached the file via device-to-device (D2D) communications, which could largely enhance the UAV endurance. The UAV was deployed in the WSN to gather environmental data for monitoring area [18]. The UAV also plays an active role in optimizing the WSN topology. The authors in [19] further considered to recharge out-of-battery ground devices with energy transfer from the UAV to extend the network lifetime. The system power consumption is minimized by deploying the UAV as a relay between the base station and sensors in [20].

It is worthwhile to note that most of the existing works mentioned above focused on energy efficiency or throughput enhancement, and only considered the UAV trajectory optimization, but overlooked the SNs transmission policy design. In fact, most of the sensory data are delay-tolerant. This character helps to further enhance the UAV data collection performance in energy efficiency and throughput, since the sensory data only needed to be uploaded to the data collector before the given time expires. Each SN needs to upload an amount of data to the data collector within a given time. At each slot, the SNs decide to upload data with optional transmission modes: i.e., (1) waiting, (2) directly transmitting data to the sink node, (3) uploading data to UAV transmission when available. On one side, the direct sink node transmission approach is always available. It nevertheless costs more energy consumption due to the distance and poor transmission link. On the other side, the UAV data gathering is energy-effective transmission mode with the short-distance LoS link, while it may not always be available. As the sensory data transmission is delay-tolerant, it is of the SN’s interest to wait for the UAV collecting data, which save transmission energy of SNs. However, constrained by the buffer size and transmission time, the long wait nevertheless leads to the unfinished transmission and incurs a buffer overflow and new sensory data loss.

In this paper, we study the delay-tolerant sensory data gathering problem in UAV-aided WSNs. Both SN’s transmission policy and UAV trajectory optimization are considered to minimize the transmission energy consumption while guaranteeing the completed transmission within a given time. We take a two-step iterative approach, and decouple the SN’s transmission design and UAV trajectory optimization process. First, we design the optimal SNs transmission mode policy with preplanned UAV trajectory. As they are aware of the given time of transmission and UAV trajectory, each SN dynamically chooses three possible modes, including (1) waiting, (2) transmitting data to sink node, (3) uploading data to UAV when possible. The multi-slot transmission mode decision can be formulated as a finite-horizon sequential Markov process [21]. A dynamic programming (DP) algorithm is proposed to obtain the optimal transmission policy. Secondly, we fix the transmission policy, and optimize the UAV’s trajectory from the preplanned trace with recursive random search (RRS) algorithm [22,23]. Numerical results show that the proposed scheme achieves significant energy saving gain over the benchmark schemes, which either ignore the transmission design or UAV trajectory optimization. It shows that the proposed scheme can strike a good trade-off between energy consumption reduction and buffer overflow avoidance.

The rest of the paper is organized as follows. The UAV-aided WSN system model is described in Section 2. The joint SN’s transmission policy and UAV trajectory optimization problem are formulated in Section 3, and the proposed algorithm in Section 4 gives the solution. Numerical results are shown in Section 5, and conclusions are drawn in Section 6.

## 2. System Model

As shown in Figure 1, we consider a wireless sensor network, where *N* sensor nodes (SN) are randomly distributed for sensing ambient environments. Each sensor node is labelled in the set N={1,2,⋯,N}, with coordinates captured in the set $A=\{a_{1},a_{2},\cdots ,a_{N}\}$, where $a_{n}\in R^{2\times 1},n\in N$. Assume $S_{n}$ bits sensory data in the *n*-th SN, n∈N needs to be delivered to the data centre within *T* seconds. The sensor nodes are usually inexpensive devices with limited buffer size. If not complete the transmission before the given time, it may result in a buffer overflow result and new sensing data lost. The given time *T* is the deadline for transmission. The total time is divided into discrete time slots, and t∈T={1,2,⋯,T}, with slot length Δt. We denote the size of remaining data in the *n*-th SN node’s buffer as $s_{n}\in S_{n}\subseteq \left[0,S_{n}\right]$.

In conventional WSNs, the *n*-th SN node sends its sensory data to the sink node through the uplink channel. Due to the long distance and huge path loss, this transmission usually comes with high energy-consuming. In this paper, we additionally deploy the UAV for sensory data gathering. The flexibility of UAV enables the shorter distance to SNs, and reduces the SNs’ transmission power consumption. Since UAV transmission is not always available for all SNs, we also consider the delay-tolerant data gathering within the transmission deadline. The SNs can keep idle and wait for the UAV coming for gathering. Thus, the SNs may have three transmission modes k∈K at each decision time:
Waiting k=0: the SN node chooses to sleep and do not transmit the sensory data;Sink transmission k=1: the SN node uploads the sensing data to the closest sink nodes;UAV gathering k=2: the SN node delivers the data to the UAV when possible.

We assume that the one fixed-wing UAV is deployed, flying at fixed *L* meters in elevation at a constant velocity *V*. The coordinates of the UAV at time slot *t* is denoted by $q_{t}$, and we can assume the UAV’s position is approximately unchanged within the slot length Δt, i.e., $\Delta t\ll \frac{1}{V}$. Therefore, the UAV’s horizontal trajectory within *T* can be approximated by the sequence $Q=\{q_{t},t\in T\}$. It satisfies that the $∥q_{T}-q_{1}∥\leq VT$. The UAV sends hello messages containing its flight information to SNs, including trajectory Q and cruise speed, at a periodic rate that is inversely proportional to its speed. Therefore, the SNs can dynamically decide the transmission policy.

The achievable data rates and transmitting power of SNs for choosing option *k* at time slot *t* are denoted by $R_{n}\left(q_{t},k\right)$ and $p_{k,t}$, respectively. It is assumed that all SNs transmit data to the sink node at orthogonal uplink subchannels with the same bandwidth $W_{1}$. From the free-space path loss model [24], we give the achievable rate of the *n*-th SN node received from the nearest sink $d_{n}$ as
(1)$$R_{n}\left(1\right)=R_{n}\left(q_{t},1\right)=W_{1}\log_{2}\left(1+\frac{p_{1,t}\kappa_{0}}{{∥b-a_{n}∥}^{\alpha }}\right),$$
where b denotes the horizontal coordinates of the sink node $d_{n}$, α is the path loss coefficient. $N_{0}$ is the average noise power. $\kappa_{0}=\frac{\beta_{0}}{N_{0}}$ is the reference received signal-to-noise (SNR), where $\beta_{0}$ represents the reference channel gain of the receiver over 1 m distance. Similarly, the achievable rates of the *n*-th SN which decides to upload data through the UAV at time slot *t* is given by
(2)$$R_{n}\left(q_{t},2\right)=W_{2}\log_{2}\left(1+\frac{p_{2,t}\kappa_{0}}{{∥q_{t}-a_{n}∥}^{\alpha /2}+L^{\alpha /2}}\right).$$
where $W_{2}$ is bandwidth for uploading data to the UAV.

## 3. Problem Formulation

In this paper, we jointly optimize the trajectory of the UAV and the transmission policy of the SNs. The energy consumption of the *n*-th SN is based on the delivered data amount and the transmission energy consumption rate of the selected mode at the time slot *t*, as given by
(3)$$E_{n,t}\left(s_{n},q,k\right)=\min\left\{s_{n},R_{n}\left(q_{t},k\right)\Delta t\right\}\eta_{n}\left(q_{t},k\right).$$
where $\eta_{n}\left(q_{t},k\right)$ is the transmission energy consumption rate (Joule/bit) [25] of the *n*-th SN node selecting the transmission mode *k*. Particularly, as no data transmitted when the SNs choose to idle, transmission energy consumption rate $\eta (q_{t},0)=0$ and the achievable data rates $R(q_{t},0)=0$.

Constrained by the limited buffer size, we define the penalty for not finishing the transmission and data loss within the given time *T*. The penalty at the time slot T+1 is expressed as
(4)$${\tilde{E}}_{n,T+1}\left(s_{n},q_{T},k\right)=\vartheta \left(s_{n}\right),$$
where the penalty $\vartheta (s_{n})$ is the function of the unfinished data at T+1, which is a non-negative function with $\vartheta (s_{n})\geq 0$ and ϑ(0)=0. Therefore, the penalty is non-decreasing with the remaining data $s_{n}$. Note that the remaining data at the end of the deadline (i.e., *T*) is bound to lose due to a buffer overflow, i.e., the remaining data at *T* is the lost data at the transmission.

We define the transmission policy as $\Omega =\{\Omega_{n},\forall n\in N\}$, where $\Omega_{n}=\left\{\alpha_{n,t}=k,\forall s_{n}\in S_{n},t\in T\right\}$ is the transmission policy for the *n*-th SN node, and $\alpha_{n,t}$ is the mode the *n*-th SN node that can take at time slot *t*. In this paper, we optimize the UAV’s trajectory and SNs transmission policy to minimize the total transmission energy consumption while minimizing data loss. The optimization problem can be formulated as
(5)$$\begin{array}{ccc} \left(P1\right): & \min_{\Omega ,Q}\;\sum_{n=1}^{N}\sum_{t=1}^{T} & E_{n,t}\left(s_{n},q_{t},\alpha_{n,t}^{\Omega }\right)+{\tilde{E}}_{n,T+1}\left(s_{n},q_{T},\alpha_{n,t}^{\Omega }\right)\\ & \;s.t. & \left(1),(2\right)\\ & & \alpha_{n,t}\in K=\left\{0,1,2\right\}\\ & & s_{n}\in S_{n}\subseteq \left[0,S_{n}\right]\\ & & ∥q_{t}-q_{t-1}∥\leq V\Delta t\\ & & a_{n}\in \left\{a_{1},a_{2},\cdots ,a_{N}\right\} \end{array}$$
where the total energy consumption and the penalty for buffer overflow within the given time is defined as the total cost function. The first part of objective of P1 is energy consumption for uploading the data from t=1 to t=T, and the second part is the penalty if the data loss occurs at t=T+1. The first one is to reduce energy consumption as much as possible, while the second aims at avoiding the data loss. There is a weight between the first and second cost function when combined. For simplicity, we assume that energy consumption and the penalty for data loss have equal weights in the cost. The weight embodies the SN’s sensitivity to energy consumption or completion. The location information of SN node $\left\{a_{1},a_{2},\cdots ,a_{N}\right\}$ is the configuration parameter predesigned at the setup time. The position of sensors are known obtained from the previous cruises of UAV data gathering. By minimizing the total cost function, we can obtain the optimal UAV’s trajectory $Q^{*}$ and SNs transmission policy $\Omega^{*}$.

## 4. Proposed Solution

Problem (P1) is a mixed-integer non-convex problem owing to the constraints (1) and (2), and it is hard to obtain the optimal solutions in general. To this end, we decouple the optimal UAV trajectory and the SNs transmission policy design process, and use a two-step iterative approach to solve P1. In the first step, we obtain the optimal SNs transmission mode policy with preplanned UAV trajectory. Then, under the given transmission policy, we optimize the UAV’s trajectory with a recursive random search (RRS) algorithm [22,23].

### 4.1. Problem Approximation

First, we simplify the optimization problem, and assume that the communication between the *n*-th SN and the UAV is available only when its data rate satisfies the threshold condition, i.e., the UAV data gathering mode is available only when it’s distance with the *n*-th SN satisfies the condition $C_{n}$, where $C_{n}\in \left\{C_{1},\cdots ,C_{N}\right\}$. Let $h_{n,t}$ denote the indicator for whether the UAV is available to the *n*-th SN at time *t*, and
(6)$$h_{n,t}=h_{n,t}\left(q_{t}\right)=\left\{\begin{array}{c} 1,\;\mathrm{if}\;∥q_{t}-a_{n}∥\leq C_{n};\\ 0,\;\mathrm{otherwise}, \end{array}\right.$$
where $h_{n,t}=1$ indicates the UAV is available for the *n*-th SN node at time *t*. For the given UAV’s trajectory Q, the UAV availability to the *n*-th SN $h_{n}$ can be mapped as the set: $H_{n}=\left\{h_{n,0},\cdots ,h_{n,t},\cdots ,h_{n,T}\right\}$. For all SNs in WSN, the availability of UAV is described as $H=\{H_{1},\cdots ,H_{N}\}$. $H_{n}^{\left(1\right)}\subseteq H_{n}$ and $H_{n}^{\left(0\right)}\subseteq H_{n}$ denote the sets of the states where UAV is available or unavailable to the *n*-th SN node or not respectively, where $H_{n}^{\left(1\right)}=H_{n}\backslash H_{n}^{\left(0\right)}$.

As a result, the mode k=2 is optional if and only if the UAV is available, i.e., $h_{n}\in H^{\left(1\right)}$. Thus, we have the *n*-th SN transmission mode set as
(7)$$K_{n}^{\left(h\right)}=\left\{\begin{array}{cc} \{0,1\}, & \mathrm{if}\;h_{n}\in H_{n}^{\left(0\right)},\\ \{0,1,2\}, & \mathrm{if}\;h_{n}\in H_{n}^{\left(1\right)}. \end{array}\right.$$

We denote the *n*-th SN node state at the time slot *t* as $w_{n,t}=\left(s_{n.t},h_{n,t}\right)$, where $s_{n,t}\in S_{n}\subseteq \left[0,S_{n}\right]$ is the size of the remaining data inside the *n*-th SN node’s buffer, and $h_{n,t}\in H_{n}$ represents the availability state of UAV to the *n*-th SN node at *t*.

With these observations, the SN transmission policy is written as $\bar{\Omega }=\left\{{\bar{\Omega }}_{n},\forall n\in N\right\}$, where ${\bar{\Omega }}_{n}=\left\{\alpha_{n,t}\left(s_{n},h_{n}\right)=k,\forall s_{n}\in S_{n},h_{n}\in H_{n},t\in T\right\}$ is the transmission policy for the *n*-th SN node. Thus, the optimization problem (P1) can be modified as
(8)$$\begin{array}{ccc} \left(P2\right):\; & \min_{\bar{\Omega },Q}\;\sum_{n=1}^{N}\sum_{t=1}^{T} & E_{n,t}\left(w_{n,t}^{\bar{\Omega }},\alpha_{n,t}^{\bar{\Omega }}\right)+{\tilde{E}}_{n,T+1}\left(w_{n,T+1}^{\bar{\Omega }},\alpha_{n,t}^{\bar{\Omega }}\right)\\ & \;s.t. & \left(1),(2),(6\right)\\ & & \alpha_{n,t}\in K_{n}^{\left(h\right)}\\ & & ∥q_{t}-q_{t-1}∥\leq V\Delta t\\ & & a_{n}\in \left\{a_{1},a_{2},\cdots ,a_{N}\right\}\\ & & w_{n,0}=\left(S_{n},h_{n,0}\right) \end{array}$$
where, $w_{n,t}^{\bar{\Omega }}=\left(s_{n}^{\bar{\Omega }},h_{n}^{\bar{\Omega }}\right)$ denotes the state when the policy $\bar{\Omega }$ is chosen at time slot *t*. $w_{n,0}$ is the initial state of the *n*-th SN node, ∀n∈N, where the $h_{n,0}$ is the availability of UAV when the UAV start from the location $q_{I}$.

Note that problem (P2) is still non-convex, so we decouple the process of the optimal UAV trajectory and the SNs transmission policy design, and so the algorithm is iterative but cannot guarantee the global optimality.

### 4.2. Optimal Transmission Policy Design with Fixed Trajectory

In this subsection, we optimize the optimal SN node transmission policy $\bar{\Omega }$ with the preplanned UAV trajectory Q. The location information of the SN node $\left\{a_{1},a_{2},\cdots ,a_{N}\right\}$ is the configuration parameter predesigned at the setup time. Thus, the availability of the UAV to the *n*-th $H_{n}$ can be derived from their geometrical mapping relation at each time slot. The SNs can decide the multi-slot decision with the complete information about the availability of UAV before their transmission deadline expires. The transmission policy ${\bar{\Omega }}_{n}$ optimization can be formulated as a finite-horizon sequential decision problem [25,26,27], as given by
(9)$$\begin{array}{ccc} \left(P3\right):\; & \min_{{\bar{\Omega }}_{n}}\;\sum_{t=1}^{T}\; & E_{n,t}\left(w_{n,t}^{{\bar{\Omega }}_{n}},\alpha_{n,t}^{\bar{\Omega }}\right)+{\tilde{E}}_{n,T+1}\left(w_{n,T+1}^{{\bar{\Omega }}_{n}},\alpha_{n,t}^{\bar{\Omega }}\right)\\ & \;s.t.\; & \alpha_{n,t}\in K_{n}^{\left(h\right)}\\ & & w_{n,0}=\left(S_{n},h_{n,0}\right). \end{array}$$

The *n*-th SN node state transition probability is defined as the probability of the state $w_{n}=\left(s_{n},h_{n}\right)$ transiting to $w_{n}^{\prime }=\left(s_{n}^{\prime },h_{n}^{\prime }\right)$ while taking mode *k* at the state $w_{n}$, i.e.,
(10)$$\begin{array}{cc} p\left(w_{n}^{\prime }|\left(w_{n},k\right)\right) & =p\left(\left(s_{n}^{\prime },h_{n}^{\prime }\right)|\left(s_{n},h_{n}\right),k\right)\\ & =p\left(h_{n}^{\prime }|h_{n}\right)p\left(s_{n}^{\prime }|\left((s_{n},h_{n}),k\right)\right), \end{array}$$
where
(11)$$p\left(s_{n}^{\prime }|\left(s_{n},h_{n}\right),k\right)=\;\left\{\begin{array}{c} 1\;\mathrm{if}\;s_{n}^{\prime }={\left[s_{n}-R_{n}\left(k\right)\Delta t\right]}^{+}\\ 0\;\mathrm{otherwise}. \end{array}\right.$$

We define $\gamma_{t}\left(w_{n}\right)$ as the optimality Bellman equation [21] of the objection function in (Equation 10) at time slot *t*, i.e., the minimal total cost for the *n*-th SN node from time slot *t* to T+1 when the node state is $w_{n,t}$ before the decision at time slot *t*,
(12)$$\gamma_{t}\left(w_{n}\right)=\min_{k\in K_{n}^{\left(h\right)}}\delta_{t}\left(s_{n},h_{n},k\right)$$
where,
(13)$$\begin{array}{c} \;\delta_{t}\left(s_{n},h_{n},k\right)\\ =E_{n,t}\left(s_{n},h_{n},k\right)+\sum_{h_{n}^{\prime }\in H_{n}}\sum_{s_{n}^{\prime }\in S_{n}}p\left(\left(s_{n}^{\prime },h_{n}^{\prime }\right)|\left(s_{n},h_{n}\right),k\right)\gamma_{t+1}\left(s_{n}^{\prime },h_{n}^{\prime }\right)\\ =\min\left\{s_{n},R_{n}\left(k\right)\Delta t\right\}\eta_{n}\left(h_{n},k\right)+\sum_{h_{n}^{\prime }\in H_{n}}\;p\left(h_{n}^{\prime }|h_{n}\right)\gamma_{t+1}\left(\;{\left[s_{n}-R_{n}\left(k\right)\Delta t\right]}^{+},h_{n}^{\prime }\right). \end{array}$$

From above, the total cost from *t* to T+1 is divided into two parts: (i) $E_{n,t}\left(s_{n},h_{n},k\right)$ is energy consumption for data transmission when the option *k* is selected at time slot *t*; (ii) the second part is the expected cost for choosing *k*, including energy consumption from t+1 to *T* and possible penalty at t=T+1 as
(14)$$\gamma_{T+1}\left(w_{n}\right)={\tilde{E}}_{n,T+1}\left(s_{n},h_{n}\right)=\vartheta \left(s_{n}\right).$$

**Theorem** **1.**
${\bar{\Omega }}_{n}^{*}=\left\{\alpha_{n,t}^{*}\left(s_{n},h_{n}\right),\forall s_{n}\in S_{n},h_{n}\in H_{n},t\in T\right\}$
*is the optimal transmission policy for the n-th SN node only when*
(15)$$\alpha_{n,t}^{*}\left(s_{n},h_{n}\right):=\underset{k\in K^{\left(h\right)}}{\arg\min}\left\{\delta_{t}\left(s_{n},h_{n},k\right)\right\},$$


**Proof.** See the principle of optimality [28]. □

With the principle of optimality and the optimality Equation (Equation 12), the optimal solution of problem (Equation 8) can be obtained based on dynamic programming. Give the granularity of $s_{n}$ as ν, such as 1 bit. According to [28], the backward induction is adopted to get the optimal transmission policy ${\bar{\Omega }}_{n}^{*}$. To be more specific, the $\gamma_{T+1}\left(w_{n}\right)$ is first set as the boundary condition. Then, obtain the $\alpha_{n,t}^{*}\left(s_{n},h_{n}\right)$ and $\gamma_{T+1}\left(w_{n}\right)$ by updating them recursively backwards with Equations (Equation 12) and (Equation 13) from t=T to time slot t=1. Under the given UAV trajectory, we illustrate the optimal transmission policy design in Algorithm 1. The details about dynamic programming algorithm can be found in [28,29].

**Algorithm 1** Dynamic programming for problem P3.
1:**Input**: Q, $S_{n}$, *T*2:Set ${\tilde{E}}_{n,T+1}\left(s_{n},h_{n}\right)$, for ∀t∈T, $\forall h_{n}\in H_{n}$ as (Equation 4)3:Set t:=T and begin in recursive backward4:
**while**
t>1
**do**
5:    **for**
$h_{n}\in H_{n}$
**do**6:        $s_{n}:=0$7:        **while**
$s_{n}<S_{n}$
**do**8:           Calculate $\delta_{t}\left(s_{n},h_{n},k\right)$, $k\in K^{\left(h\right)}$ using (Equation 13)9:           Set $\alpha_{t}^{*}\left(s_{n},h_{n}\right):=\underset{k\in K^{\left(h\right)}}{\arg\min}\left\{\delta_{t}\left(s,h,k\right)\right\}$10:           $\gamma_{t}\left(s_{n},h_{n}\right)=\delta_{t}\left(s_{n},h_{n},\alpha_{t}^{*}\left(s_{n},h_{n}\right)\right)$11:           s:=s+ν12:        **end while**13:    **end for**14:    t:=t−115:
**end while**
16:**Output**: the optimal strategy $\Omega_{n}^{*}$.


### 4.3. UAV Trajectory Optimization with Fixed SNs Transmission Policy

In this subsection, we optimize the UAV trajectory with the fixed transmission policy $\bar{\Omega }$. The UAV trajectory optimization problem is formulated as:
(16)$$\begin{array}{ccc} \left(P4\right): & \min_{Q}\sum_{n=1}^{N}\sum_{t=1}^{T} & E_{n,t}\left(w_{n,t}^{\bar{\Omega }},\alpha_{n,t}^{\bar{\Omega }}\right)+{\tilde{E}}_{n,T+1}\left(w_{n,T+1}^{\bar{\Omega }},\alpha_{n,t}^{\bar{\Omega }}\right)\\ & \;s.t. & \left(1),(2),(6\right)\\ & & ∥q_{t}-q_{t-1}∥\leq V\Delta t\\ & & a_{n}\in \left\{a_{1},a_{2},\cdots ,a_{N}\right\}. \end{array}$$

The problem is still non-convex even with fixed SNs transmission policy. Therefore, we propose to find the optimal trajectory of the UAV based on Recursive Random Search (RRS) algorithm [22,23,30]. The algorithm can search the close-to-optimal solution from the preplanned trajectory $\bar{Q}$. For simplicity, the preplanned trajectory $\bar{Q}$ is assumed as a circle with the center at the sink node, where UAV can fly pass or close the maximum number of SNs. At each iteration *i*, initializes *Z* position candidates $q_{t}^{z}$, z=1,⋯,Z around the UAV position $q_{t}$, and these candidates satisfy the UAV trajectory constraints. Then, find the best candidate position $q_{t,i}^{*}$ at the *i*-th iteration that maximizes the objective function in (Equation 16). After that, we continue to apply the shrink-and-realign searching process to obtain the best candidates $q_{t}^{*}$ among all iteration results $q_{t,i}^{*}$. The shrink-and-realign procedure stop when the threshold condition satisfies. We illustrate the shrink-and-realign procedure in Algorithm 2. The details can be found in [22,30].

**Algorithm 2** Recursive random search algorithm for P4.
1:**Input**: preplanned trajectory $\bar{Q}$,2:
**for**
t=1:T
**do**
3:    i←0; set tolerance ϵ>0;4:    Generate position candidates $q_{t}^{z}$, z=1⋯Z5:    **repeat:**6:    **for**
z=1:Z
**do**7:        Find $q_{t,i}^{*}=\arg\min_{q_{t,i}^{z}}\sum_{n=1}^{N}\sum_{t=1}^{T}E_{n,t}+{\tilde{E}}_{n,T+1}$;8:    **end for**9:    l←l+1;10:  Find $q_{t}^{*}$ with shrink-and-realign sampling process;11:  **until** fractional increase of the objective value of (P4) is below ϵ.12:
**end for**



## 5. Numerical Results

In this section, we will evaluate the effectiveness of the proposed method for UAV-aided WSN by simulation. We considered a single UAV deployed in the WSN for data gathering, flying at a fixed altitude L=50 m. As shown in Figure 2, we considered the WSN with one sink node and nine SNs in total, where the sink node was located at the center of the area, and SNs are distributed randomly within an area of 400 m × 400 m. Notice that, we only considered a single sink node situation in the simulation, but the proposed algorithm can be easily extended to the scenario with multiple sink nodes. Simulation parameters were set as follows: $N_{0}=-80$ dBm, subchannel bandwidth $W_{1}=W_{2}=1$ MHz, transmit power $p_{1,t}=p_{1}=30$ dBm and $p_{2,t}=p_{2}=25$ dBm. The channel power gain at reference distance $\beta_{0}=-20$ dB and the path loss coefficient α=2. We fit the energy consumption rate as an exponential function [25] of the achievable data rate from the sample data [31]. The energy consumption rates for sink node transmission is $\eta \left(R_{1,t}\right)=1.42*e^{\left(-0.0093*R_{1,t}\right)}$, and for UAV data gathering approach is $\eta \left(R_{2,t}\right)=1.4*e^{\left(-0.053*R_{2,t}\right)}$. The tolerance for RRS algorithm enabling the UAV is assumed as ±40 m. The penalty function for data loss is $\vartheta \left(s\right)=20s^{2}$ [26]. The results were obtained over 1000 times simulations with randomized SNs locations.

We examine the proposed trajectory method based on the RSS algorithm with ±40 m tolerance in Figure 2. With the RSS method in [30], it is observed that the proposed algorithm has more freedom to modify the UAV’s trajectory from the preplanned trajectory. The tolerance in RSS algorithm enables the UAV to explore closer positions to the SNs from the original ones, and thus enhances the data connectivity and throughput. The close position increases the possibility of the LoS connection between UAV and SNs, and lower energy consumption.

We further demonstrate the energy efficiency performance of the proposed algorithm. The energy efficiency. is defined as energy consumption per Mbit (Joule/Mb) in this paper. [25]. The following three benchmarks are considered for comparison:
Optimal transmission scheme: in this scheme, only the SNs transmission policy is optimized based on the preplanned UAV trajectory. Specifically, the SNs obtain the deadline of transmission and the trajectory and make their decision on the transmission mode at each time slot. The optimal transmission policy design follows the dynamic programming in Algorithm 1 in Section IV-B. Similar approaches can be taken from [26,27].Optimal trajectory scheme: in this scheme, only the UAV trajectory is optimized. The UAV is not aware of the SNs transmission policy, and only optimize the trajectory based on SNs positions. The SNs choose their transmission mode in a heuristic method and try to deliver the data as soon as possible without the awareness of the transmission deadline, i.e., the *n*-th SN uploads data to the UAV when the UAV transmission is possible $h_{n}=1$, and chooses to transmit to the sink node when $h_{n}=0$. The similar schemes are used in [32,33].Sink-SN transmission scheme: in this scheme, the SNs transmit sensory only by sink node transmission approach k=1. The similar schemes can be found in [34,35].

First, Figure 3 illustrates the energy efficiency of each scheme over different transmission deadline *T*. As the increase of *T*, the energy efficiency of the proposed algorithm improves since more chances to upload data to UAV. Notice that, the transmission efficiency of SNs can be improved by the leveraging the LoS link with high channel gains. The more access to the UAV, and more energy efficiency gain can be achieved. With the trajectory optimization, the proposed algorithm can place the UAV in a better position to harvest the channel gain, and thus achieves the highest energy efficiency, compared with the optimal transmission scheme in which the UAV trajectory is not considered. As not aware of the deadline, the optimal trajectory scheme chose sink node transmission, i.e., a low energy-efficient approach, and fail to utilize the UAV data gathering fully. Sink-SN transmission scheme acted as the low bound of the energy efficiency performance as only sink node transmission is considered in this scheme. Moreover, we evaluated the performance of two kinds of SNs, i.e., near point $R_{1}=20$ Mbps and far point $R_{1}=15$ Mbps. It can be observed that the far-point SN achievement was low compared with the near one, which was caused by the serious path loss and poor channel gain of far point SN.

In Figure 4, the energy efficiency comparison under different data request $S_{n}$ is plotted. It is observed that, for given deadline T=80 s, the energy efficiency of the proposed algorithm decreases with the larger data request. It is because, facing a larger amount of data request, the proposed algorithm would select to transmit data to the sink node directly to avoid the data loss penalty. As more data is required to to be uploaded, more sink transmission modes k=1 are adopted, which is at the cost of energy consumption. In summary, the proposed algorithm can get a good tradeoff between energy consumption and the upload demand. Since the absence of optimizing the UAV trajectory, the optimal transmission scheme cannot get the larger throughput from a suboptimal position at the preplanned trajectory. Through the UAV trajectory is optimized in the optimal trajectory scheme: scheme, it chooses a heuristic method rather than fully considering the SN transmission policy together with the deadline. It is noticed that the performance of the optimal trajectory scheme act as the lower bound of the proposed algorithm when large data are requested under a short deadline.

## 6. Conclusions

In this paper, we investigate the UAV-aided wireless sensor networks system, where a UAV is deployed as auxiliary data gathering node for delay-tolerant sensor data. With the assistance of UAV data gathering, each sensor node dynamically choose the transmission modes, i.e., (1) waiting, (2) conventional sink node transmission, (3) uploading to UAV when possible, to transmit sensory data within the time tolerance. Benefit from the LoS link between the UAV and SNs, it is an incentive for the SNs to wait for the UAV’s arrival and upload data to the UAV to reduce the transmit energy consumption before the deadline expires. Aimed at minimizing the transmission energy consumption while guaranteeing the transmission completion, both SN’s transmission policy and UAV trajectory optimization problems are jointly considered in this paper. A two-step iterative approach is adopted, where the SN’s transmission policy and UAV trajectory optimization are solved iteratively. The multi-slot transmission mode decision is formulated as a finite-horizon sequential Markov process in this paper [21]. A dynamic programming (DP) algorithm is proposed to obtain the optimal transmission policy. Secondly, we fix the transmission policy, and optimize the UAV’s trajectory from the preplanned trace with RRS algorithm. Numerical results show that the proposed scheme achieves significant energy savings gain over the benchmark schemes, which either ignore the transmission design or UAV trajectory optimization. It is shown that the proposed scheme can strike a good balance between energy consumption reduction and buffer overflow avoidance. For future research, we propose to extend the model and take the limited service capacity of the UAV into consideration. It is interesting to investigate the effects of the UAV service capacity constraint on system performance.

## Figures and Tables

**Figure 1 sensors-19-02506-f001:**
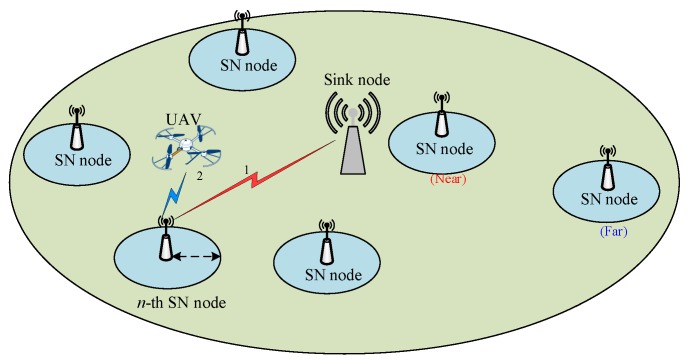
System model.

**Figure 2 sensors-19-02506-f002:**
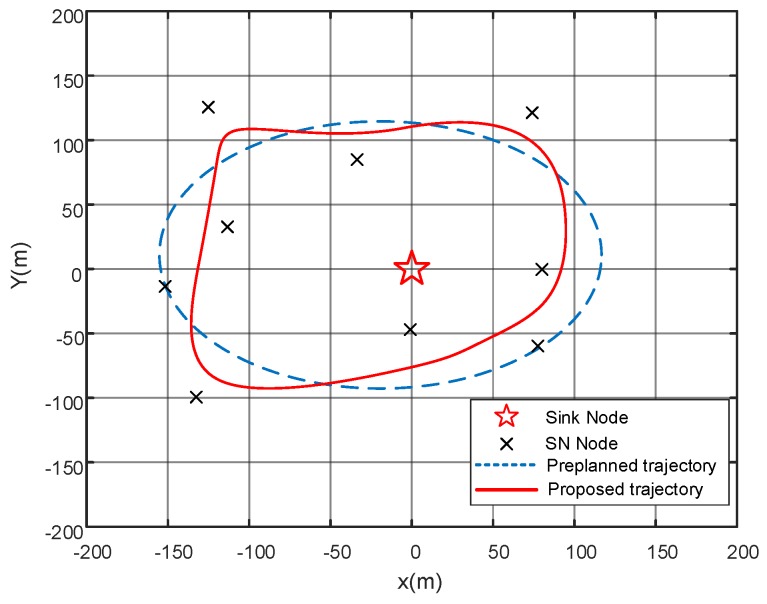
The proposed recursive random search (RRS)-based unmanned aerial vehicle (UAV) trajectory algorithm and preplanned trajectory

**Figure 3 sensors-19-02506-f003:**
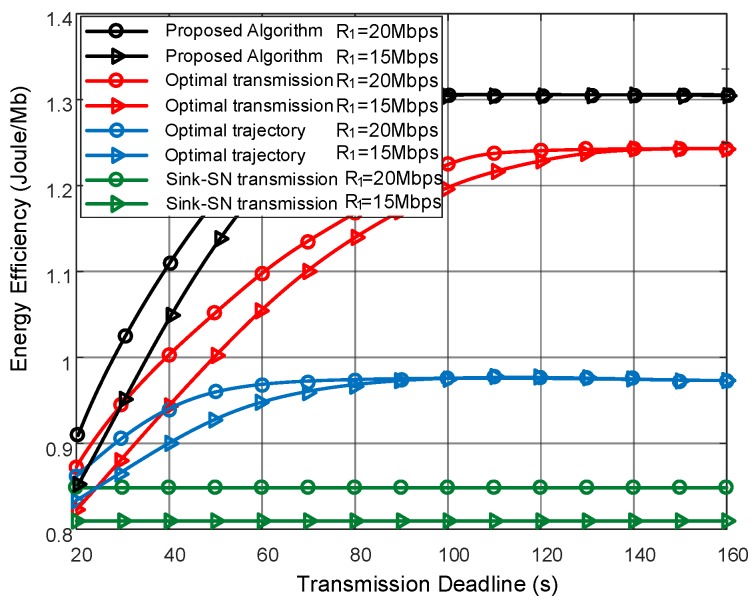
Energy efficiency comparison under different transmission deadlines with file size $S_{n}=400$ Mbits.

**Figure 4 sensors-19-02506-f004:**
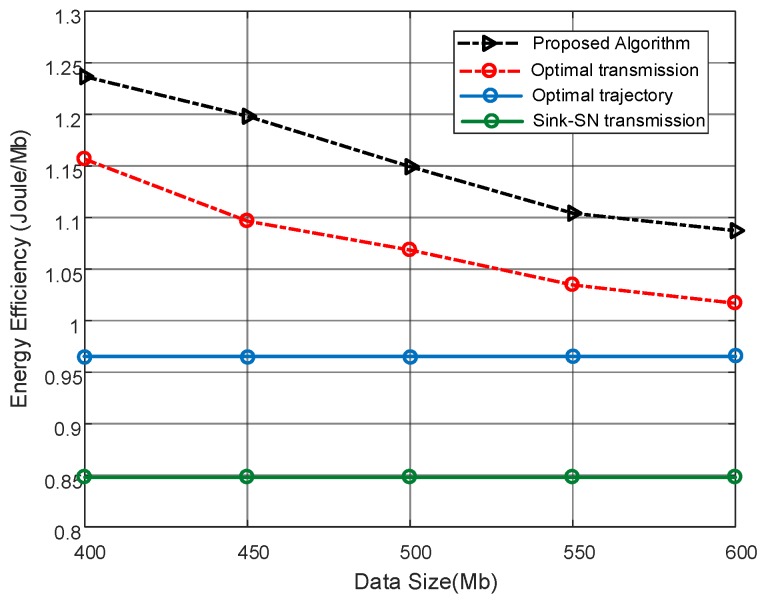
Energy efficiency comparison versus different data request within T=80 s, and $R_{1}=20$ Mbps.

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
