# Peer review of "Energy-Effective Data Gathering for UAV-Aided Wireless Sensor Networks"

_sensors, 2019, doi:10.3390/s19112506_

Round 1
Reviewer 1 Report
The authors present a detailed paper in which different mathematical models are presented that allow to determine the trajectory of a UAV for data collection in sensor networks, with the aim of reducing energy consumption.
One of the possible improvements of the paper is to include a more detailed related work, for example, here I include some recent papers that could be cited:
- Popescu, D., Dragana, C., Stoican, F., Ichim, L., & Stamatescu, G. (2018). A collaborative UAV-WSN network for monitoring large areas. Sensors, 18(12), 4202.
- Arabi, S., Sabir, E., Elbiaze, H., & Sadik, M. (2018). Data Gathering and Energy Transfer Dilemma in UAV-Assisted Flying Access Network for IoT. Sensors, 18(5), 1519.
- Fu, S., Zhao, L., Su, Z., & Jian, X. (2018). UAV based relay for wireless sensor networks in 5G systems. Sensors, 18(8), 2413.
- Lan, K. C., & Wei, M. Z. (2017). A compressibility-based clustering algorithm for hierarchical compressive data gathering. IEEE Sensors Journal, 17(8), 2550-2562.
- Gharaei, N., Bakar, K. A., Hashim, S. Z. M., Pourasl, A. H., & Butt, S. A. (2018). Collaborative mobile sink sojourn time optimization scheme for cluster-based wireless sensor networks. IEEE Sensors Journal, 18(16), 6669-6676.
- Sharma, V., You, I., Pau, G., Collotta, M., Lim, J., & Kim, J. (2018). Lorawan-based energy-efficient surveillance by drones for intelligent transportation systems. Energies, 11(3), 573.
On the other hand, it is necessary to determine with more precision how the pre-planned trajectory of the drone is obtained, in particular, how this trajectory is obtained. Also, how is the position of the sensors known?
As for the results, a table with the necessary values must be included to make the proposal reproducible, such as:
- number of nodes
- location of the sink node
- initial energies of the sensors
- number of simulations
- other parameters included in the model
Also, how much time is necessary for the iterative algorithms to converge? Has an estimate been made of this time? What is the influence of this time on the proposed algorithm?
Finally, is it possible to compare with other authors' algorithms?
Author Response
Thank you very much for your letter dated 7 May 2019, about our paper sensors-501587 titled “Energy-effective Data Gathering for UAV-Aided Wireless Sensor Networks” as well as for your comments and those from the reviewers. We have checked and revised the manuscript in accordance with your and the reviewers’ comments and have carefully proofread the manuscript to minimize typographical, grammatical, and bibliographical errors. Please find below our description of the revisions accordingly in the attachment.

Reviewer 2 Report
This paper presents a joint optimization scheme for SN transmission policy and UAV trajectory in wireless sensory networks. This paper chooses a scenario where no previous literature didn't fully cover. There are some issues that need to be addressed.
The system model is for multiple sinks, yet the simulation is only conducted for a single sink. Please explain why.
Please elaborate on the choice of the penalty function for data loss.
In (P1) two cost functions are combined into one. Please elaborate on considerations to be taken when doing so.
The formulation does not consider the UAV's capability to receive uploads from multiple SNs simultaneously. Is it not an important issue?
Apparently the initial trajectory has great effects on final results in the 2nd step of optimization. Please elaborate on that.
Comparison to three so-called "benchmark" schemes does not justify much. All three benchmarks are subsets of the proposed scheme. Not much can be said from the comparison. Can some schemes from the survey literature be modified for the simulation setting and compared to the proposed scheme?
Please provide some justifications for the simulation settings. For instance, give an application scenario where the setting seems typical.
The writing can be improved.
SN nodes are often used while SN is enough.
The unit for energy efficiency in Fig. 3 and 4 is wrong.
Please settle the name for the search algorithm in the 2nd step (RRS or RUS?).
I have highlighted in the attached file the places where grammar errors should be corrected or better choice of words should be exercised.

Author Response

(The authors gave the same response as above.)

Round 2
Reviewer 1 Report
The authors have made all the changes appropriately. The paper has been improved and, from my point of view, no further changes are necessary.
Author Response
Thank you very much for your letter dated 26 May 2019, about our paper sensors-501587 titled “Energy-effective Data Gathering for UAV-Aided Wireless Sensor Networks” as well as for your comments and those from the reviewers. We have checked and revised the manuscript in accordance with your and the reviewers’ comments and have carefully proofread the manuscript to minimize typographical, grammatical, and bibliographical errors. Please find below our description of the revisions accordingly in the attachment.

Reviewer 2 Report
Most of my concerns are addressed properly. But I would like the authors to clearly mentioned the limitations of their approach. Especially, as I said in previous reviews, the multiuser ability of the UAV receiver and the trade-off between the two goals in the cost function, are not explicitly treated in this paper.
Author Response
We are very grateful for your comments of our manuscript. Many grammatical or typographical errors have been revised. All sections and figures referred above are in the revised manuscript. We thank you and all the reviewers for all the valuable suggestions. We would gladly appreciate it if you would not hesitate to let us know if there are more issues with regard to our manuscript and our responses.
Best regards.
Sincerely yours,
All Authors